# GATED LoRA: DUAL-PURPOSE PROJECTIONS FOR PARAMETER-EFFICIENT FINE-TUNING

## ABSTRACT

Low-Rank Adaptation (LoRA) is widely used for parameter-efficient fine-tuning (PEFT) of large language models (LLMs). Yet, its uniform activation of all rank components can lead to task interference and hinder generalization when fine-tuning a model to multiple tasks and datasets. We introduce Gated LoRA, which employs input-dependent gating to selectively activate only the most relevant rank-1 directions. The key design is a dual-purpose projection: the same matrices that compute LoRA features also drive rank selection, adding no extra trainable parameters. Across nine language understanding benchmarks and diverse LLM backbones, Gated LoRA reduces task interference and achieves up to 2.9-point accuracy gains in multi-task settings and 3.6-point gains in single-task fine-tuning over standard LoRA, while incurring negligible inference latency overhead and no additional training parameters. Our results demonstrate that fine-grained, input-dependent adaptation makes LoRA more robust, adaptive, and interference-resistant, suggesting a way for scalable multi-task PEFT in LLMs.

## 1 INTRODUCTION

Pre-trained Large Language Models (LLMs), built with billions of parameters and trained on web-scale corpora, have become the foundation for many downstream applications (Radford et al., 2019; Brown et al., 2020; Touvron et al., 2023a). Yet, fully fine-tuning these models to each new task or domain remains challenging, as it is computationally expensive and requires significant memory. Parameter-efficient fine-tuning (PEFT) (Houlsby et al., 2019) offers a practical alternative by restricting updates to a small subset of parameters. Among PEFT methods, Low-Rank Adaptation (LoRA) (Hu et al., 2022) stands out for its balance of simplicity and effectiveness, which constrains updates to a low-dimensional subspace via low-rank matrix factorization.

Despite their success, conventional LoRA faces critical limitations in multi-task settings, scenarios that are essential for efficiently training a single generalist model. When adapting a model to multiple diverse tasks, LoRA applies all rank components uniformly to every input, thereby ignoring task-specific variations. This lack of selectivity can cause task interference, whereby updates that benefit one task may hinder another, and also lead to suboptimal parameter utilization as all rank directions are equally active. The problem becomes more pronounced as downstream tasks grow more diverse and complex (Liu et al., 2024; Zhao et al., 2025a; Feng et al., 2024; Tian et al., 2024; Dou et al., 2024; Qing et al., 2024).

One natural way to introduce input-dependent selectivity is through conditional computation, as explored in Mixture-of-Experts (MoE) architectures (Jacobs et al., 1991). Recent PEFT variants pair LoRA with MoE-inspired sparse selection over multiple adapters, often by attaching multiple LoRA modules and training a separate routing network to choose among them (Dou et al., 2024; Gao et al., 2024; Feng et al., 2024; Qing et al., 2024; Zadouri et al., 2024). While effective, these designs introduce additional routing parameters and training complexity, and their coarse, block-wise activation can miss fine-grained input variation (Ludziejewski et al., 2024).

In this paper, we introduce Gated LoRA, a LoRA variant inspired by the principles of conditional computation, but with a self-gated and parameter-shared architecture that performs rank-level selection within a single adapter. Instead of adding external experts or a separate router, we reinterpret each rank-1 LoRA component as a fine-grained feature direction and introduce an input-dependent

selector that activates only the most relevant ranks per input. Crucially, the same low-rank projection used by LoRA to compute features is also reused to produce selection scores, creating a dual-purpose design that adds no extra trainable parameters beyond standard LoRA while enabling fine-grained, input-adaptive updates.

We study two selectors: (i) a fixed top-$k$ activation that selects the $k$ strongest rank directions by magnitude, and (ii) a dynamic ReLU-based activation regularized for sparsity, which allows the number of active ranks to vary by input. To reduce interference and promote rank-wise specialization, we further apply an orthogonality regularizer over the up-projection directions. Together, these choices yield a simple, end-to-end trainable module that preserves LoRA's efficiency while improving its adaptability.

Empirically, across nine language-understanding benchmarks and diverse backbones (including LLaMA, Phi-3, and Qwen-2), Gated LoRA improves average accuracy by up to +2.9 percentage points in multi-task training and +3.6 in single-task fine-tuning over standard LoRA, under the same PEFT budget. The method maintains LoRA-level training practicality and incurs negligible inference-time overhead relative to LoRA. We also provide ablations on the selector type and regularization configurations to analyze when and why selective rank activation helps. A controlled ablation demonstrates that the dual-purpose projection is both more accurate and more parameter-efficient than adding a separate gating layer with its own parameters.

Our key contributions include:

- **Self-gated LoRA with dual-purpose projections:** a simple, parameter-shared design that reuses LoRA's projection to select active ranks, adding no extra trainable parameters.
- **Selectors and regularization for Gated LoRA:** fixed top-$k$ and sparsity-regularized ReLU selectors, plus an orthogonality loss that encourages diverse update directions and mitigates task interference.
- **Comprehensive evaluation and analysis:** gains over standard LoRA and MoE-inspired PEFT baselines on nine benchmarks and diverse model families, with ablations on sparsity, orthogonality, and $r/k$ budgets, and measurements showing negligible inference overhead.

## 2 RELATED WORK

### 2.1 PARAMETER-EFFICIENT FINE-TUNING AND LOW-RANK ADAPTATION

Fine-tuning pre-trained language models has proven effective for adapting to specific downstream tasks(Radford et al., 2019; Brown et al., 2020). However, the computational and storage overhead of full-model fine-tuning becomes prohibitive as model sizes grow. Parameter-efficient fine-tuning (PEFT) (Houlsby et al., 2019) mitigates these costs by updating only a small subset of parameters while keeping the majority of pre-trained weights frozen.

Several PEFT methods have been introduced, including adapter modules (Houlsby et al., 2019), prompt tuning (Lester et al., 2021), and prefix tuning (Li & Liang, 2021). Among them, Low-Rank Adaptation (LoRA) (Hu et al., 2022) has gained widespread adoption due to its simplicity and effectiveness. LoRA injects trainable low-rank matrices into the frozen linear base layers, allowing updates to be approximated within a low-dimensional subspace. This significantly reduces the number of trainable parameters while preserving strong performance across tasks. Additionally, LoRA's modular nature enables flexible integration across multiple tasks without modifying the base model. Despite its effectiveness in single-task scenarios, LoRA encounters limitations in multi-task settings, as the uniform application of rank components can lead to task interference and inefficient parameter utilization (Liu et al., 2024; Zhao et al., 2025a; Feng et al., 2024; Tian et al., 2024).

### 2.2 MIXTURE-OF-EXPERTS (MOE)

Mixture-of-Experts (MoE) (Jacobs et al., 1991) incorporates multiple specialized sub-networks ("experts"), along with a routing mechanism that selectively activates or combines these experts based on input characteristics. This strategy enables conditional computation, optimizing resource allocation by engaging different subsets of the network depending on the input.

Recent advancements in MoE-based large language models, such as Mixtral (Jiang et al., 2024) and Switch Transformers (Fedus et al., 2022), have demonstrated considerable efficiency and performance gains. By activating only a small number of experts per input, these models significantly reduce computational overhead while maintaining or improving the performance compared to fully dense models. This approach has been extended in more recent systems such as DeepSeek-V2 (DeepSeek-AI, 2024). In addition, recent works have also explored alternatives to the standard top-$k$ softmax routers for dynamic allocation and scalable training (Wang et al., 2025).

### 2.3 PARAMETER-EFFICIENT MoE FOR FINE-TUNING

To combine the benefits of PEFT and sparse conditional computation, recent work has explored integrating MoE architectures with parameter-efficient fine-tuning. These methods typically attach a mixture-of-experts module composed of multiple small LoRA adapters in parallel to the frozen backbone weights. A separate router network dynamically selects among these experts, enabling input-dependent adaptation with reduced training costs.

LoRAMoE (Dou et al., 2024) combines multiple LoRA modules with a learned router network to support input-dependent adaptation while preserving general knowledge from the pre-trained model. MoELoRA (Liu et al., 2024) extends this idea to multi-task medical applications, employing low-rank experts gated by task-specific signals. Other notable approaches include MoLA (Gao et al., 2024) and AlphaLoRA (Qing et al., 2024), which adjust expert composition at the layer level, and MoV (Zadouri et al., 2024), which activates expert vectors via lightweight router mechanisms for instruction tuning. While effective, these methods typically rely on coarse, block-wise expert activation and introduce additional routing parameters, which may limit their ability to capture fine-grained task variations.

## 3 GATED LoRA

We propose Gated LoRA, a self-gated extension of LoRA that introduces input-dependent rank selection without any separate routing module. Each rank-1 LoRA component is treated as a fine-grained feature direction, and the same low-rank projection that extracts features also produces selection scores, as illustrated in Figure 1. By combining the parameter efficiency of LoRA with input-dependent sparse activation over rank directions, Gated LoRA offers a scalable and effective framework for multi-task adaptation.

### 3.1 FROM LOW-RANK ADAPTATION TO RANK-WISE FEATURE DIRECTION

Let $\boldsymbol{W}_0 \in \mathbb{R}^{d_{\text{out}} \times d_{\text{in}}}$ be a frozen base layer which takes the input $x \in \mathbb{R}^{d_{\text{in}}}$ and outputs $y = \boldsymbol{W}_0 x \in \mathbb{R}^{d_{\text{out}}}$. Standard LoRA (Hu et al., 2022) parameterizes an additive low-rank update $\Delta \boldsymbol{W}$ as:

$$\Delta \boldsymbol{W} = \boldsymbol{B}\boldsymbol{A}, \ \text{where} \ \boldsymbol{B} \in \mathbb{R}^{d_{\text{out}} \times r}, \ \boldsymbol{A} \in \mathbb{R}^{r \times d_{\text{in}}}, \ r \ll \min(d_{\text{in}}, d_{\text{out}}), \tag{1}$$

which can be written as a sum of rank-1 updates:

$$\Delta \boldsymbol{W} = \sum_{i=1}^{r} b_i a_i^{\top}, \tag{2}$$

where $b_i \in \mathbb{R}^{d_{\text{out}}}$ is the $i$-th column of $\boldsymbol{B}$ and $a_i \in \mathbb{R}^{d_{\text{in}}}$ is the $i$-th row of $\boldsymbol{A}$.[1]

We interpret each pair $(a_i, b_i)$ as a rank-1 feature direction: $a_i$ detects an input feature and $b_i$ applies the corresponding output adjustment. Given input $x$, the layer output is

$$y = \boldsymbol{W}_0 x + \Delta h(x), \qquad \Delta h(x) = \boldsymbol{B}u(x), \tag{3}$$

where $u(x) \in \mathbb{R}^r$ is an input-dependent vector. For standard LoRA, this is simply a low-rank down-projection feature $u(x) = \boldsymbol{A}x$.

---

[1]In practice, we scale the LoRA update as $\Delta \boldsymbol{W} = (\alpha/r)\boldsymbol{B}\boldsymbol{A}$, where $\alpha$ is a tunable hyperparameter and $r$ is the rank.

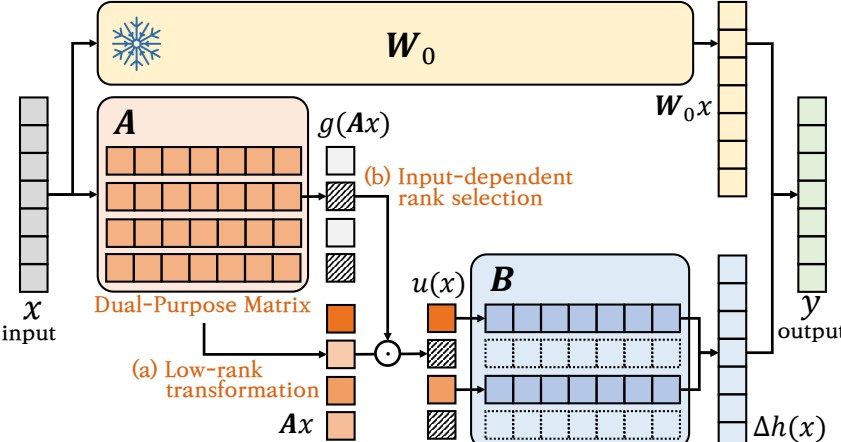

Figure 1: **Overview of Gated LoRA architecture.** The figure illustrates a Gated LoRA with $d_{\text{in}} = d_{\text{out}} = 7$, rank $r = 4$, and top-2 activation. The module is attached in parallel to a frozen base layer $\boldsymbol{W}_0$. The down-projection layer $\boldsymbol{A}$ serves a dual purpose: (a) providing low-rank features while (b) driving input-dependent rank selection simultaneously. A gating function $g$ (with no training parameters) creates a binary mask from the absolute values of the projected vector, $\boldsymbol{A}x$, which is then applied to the features to produce the sparse vector $u(x)$. This sparse vector is passed through the up-projection matrix $\boldsymbol{B}$ to compute the output update $\Delta h(x)$.

### 3.2 TOP-$k$ GATED RANK DIRECTIONS

Standard LoRA applies all rank-1 components uniformly for every input. In contrast, we introduce input-dependent gating to selectively activate the most relevant rank directions for each input:

$$\Delta h(x) = \boldsymbol{B} \cdot \text{diag}(g(\boldsymbol{A}x)) \cdot \boldsymbol{A}x, \tag{4}$$

or equivalently, $u(x) = g(\boldsymbol{A}x) \odot \boldsymbol{A}x$ where $g(\boldsymbol{A}x)$ is a gating function.

Let $s(x) = \boldsymbol{A}x \in \mathbb{R}^r$ be the low-rank features. We compute a selector $g : \mathbb{R}^r \mapsto \{0, 1\}^r$ that activates only the $k$ strongest directions by magnitude:

$$g_{\text{top-}k}(s)_i = \mathbf{1}\left[i \in \text{TopK}_k(|s|)\right], \qquad u(x) = g_{\text{top-}k}(s) \odot s. \tag{5}$$

Using the absolute value $|s|$ for selection ensures that rank-1 components are selected based on the strength of their activation, regardless of whether the signal is positive or negative. The Hadamard product with $s$ preserves the original sign, retaining the directional information of each active component. We use a straight-through estimator for $g_{\text{top-k}}$ training: the forward pass applies the hard mask, while the backward pass routes gradients through $s$.

This self-gating mechanism ultimately creates a dual role for matrix $\boldsymbol{A}$. It serves both for pattern detection, where each row $a_i$ detects specific input patterns in $x$, and for selection information, where the resulting projection $\boldsymbol{A}x$ serves as the selection signal that determines which rank directions should be activated. This integrated mechanism eliminates the need for separate routing parameters, as the same matrix $\boldsymbol{A}$ that performs low-rank adaptation also informs the gating decisions.

### 3.3 DYNAMIC ReLU SELECTOR WITH SPARSITY CONTROL

In conventional MoE implementations, it is common to fix the number of active experts $k$ per input. However, this assumption may not hold in our setting with fine-grained rank directions: the optimal number of active components can vary across tasks and even across inputs within a task (e.g., simpler inputs recruit few directions; harder ones recruit more).

To accommodate this variability, we adopt ReLU gating as an alternative to fixed top-$k$ activation. ReLU gating activates a rank component only when the selector's input is positive,

$$u(x) = \text{ReLU}(\boldsymbol{A}x), \qquad \Delta h(x) = \boldsymbol{B}u(x), \tag{6}$$

which yields a variable number of active ranks per input rather than a fixed number of top-$k$ selections. This is equivalent to having a thresholded selector $g(s) = \mathbf{1}[s > 0]$.

Because ReLU does not constrain the number of active ranks, we encourage a target activation density using $\ell_1$ norm regularization. Concretely, we utilize a hinge-style $\ell_1$ sparsity penalty applied only when the selector is denser than desired. Defining the active fraction $f(x) = \frac{1}{r}\|\text{ReLU}(\boldsymbol{A}x)\|_0$, we use a target activation ratio $\rho = k/r$ (to align with top-$k$ baselines) and penalize

$$\Phi(\boldsymbol{A}x) = \begin{cases} \|\text{ReLU}(\boldsymbol{A}x)\|_1, & f(x) > \rho \\ 0, & \text{otherwise} \end{cases}. \tag{7}$$

This supplies a steady negative gradient on active entries when over-activation occurs, encouraging the model to minimize redundant activations while retaining the flexible rank selection. Note that $f(x)$ only acts as a density check and is treated with stop-gradient.

## 3.4 ORTHOGONAL REGULARIZATION FOR RANK-WISE SPECIALIZATION

To encourage diverse rank directions and reduce interference among simultaneously active ranks, we penalize correlation among the columns of matrix $\boldsymbol{B}$:

$$\Omega(\boldsymbol{B}) = \|\boldsymbol{B}^\top \boldsymbol{B} - \boldsymbol{I}\|_F^2, \tag{8}$$

where $\boldsymbol{I} \in \mathbb{R}^{r \times r}$ is an identity matrix.

The penalty drives the columns of $\boldsymbol{B}$ toward orthogonality, so that each rank captures a distinct transformation. By reducing redundancy, it improves capacity utilization and coverage of the output space, and also limits cross-terms among simultaneously activated ranks, allowing components to contribute additively rather than interfere. Prior work (Wang et al., 2023) has applied orthogonality across task-specific adapters to mitigate forgetting in continual learning; here we apply the same principle within a single adapter, across rank directions, to reduce within-task interference.

Across adapted layers $\mathcal{L}$ and inputs $x \sim \mathcal{D}$, the full objective is:

$$\mathcal{J}(x) = \mathcal{J}_{\text{ft}}(x) + \lambda_1 \mathbb{E}_x \left[ \frac{1}{|\mathcal{L}|} \sum_{\ell \in \mathcal{L}} \Phi^{(\ell)}(\boldsymbol{A}x) \right] + \lambda_2 \frac{1}{|\mathcal{L}|} \sum_{\ell \in \mathcal{L}} \Omega^{(\ell)}(\boldsymbol{B}), \tag{9}$$

where $\mathcal{J}_{\text{ft}}(x)$ is the fine-tuning loss. $\lambda_1$ and $\lambda_2$ are hyperparameters controlling the strength of the regularization. In inference, the incremental cost relative to LoRA is a layer-local diagonal mask $(u(x))$ applied inside the low-rank branch. No extra trainable parameters are introduced.

## 4 EXPERIMENT DETAILS

In this section, we describe the experimental settings used for LLM parameter-efficient fine-tuning. We conduct two types of experiments: single-task fine-tuning and multi-task fine-tuning.

**Datasets.** We conduct the experiment based on diverse question answering datasets, including ARC (Clark et al., 2018) (both Easy and Challenge subsets), BoolQ (Clark et al., 2019), Open-BookQA (OBQA) (Mihaylov et al., 2018), PIQA (Bisk et al., 2019), SocialIQA (SIQA) (Sap et al., 2019), HellaSwag (HellaS) (Zellers et al., 2019), WinoGrande (WinoG) (Sakaguchi et al., 2021), and CommonSenseQA (CSQA) (Talmor et al., 2019). Each dataset involves different tasks such as question-answering, text classification, or fill-in-the-blank tasks, as well as different domains such as natural science, physical interactions, social interactions, or video captions. We use the first five datasets for multi-task fine-tuning. Details can be found in Appendix A.1.

**Baselines.** We compare our approach against several strong baselines: standard LoRA (Hu et al., 2022), LoRAMoE (Dou et al., 2024) as a representative of softmax-based routing, and MoLA (Gao et al., 2024) as a representative of top-$k$ routing strategies. To ensure a fair comparison, we fix the total number of ranks per adapter to 128 across all methods. For LoRAMoE and MoLA, we follow their respective configurations with 8 experts, where MoLA selects the top-2 experts per input (resulting in 32 active ranks).

Table 1: Accuracy on **multi-task** fine-tuning (five datasets) with LLaMA backbones. All models use 128 total ranks. **Params.%** is the proportion of trainable parameters. $\Delta$ is the average accuracy improvement over the LoRA baseline (percentage points). **sec/batch** is the average training/inference time per batch. Inference latency was evaluated with LLaMA-3.1 8B backbone.

| Method | Params.% | Accuracy (%) ↑ | | | | | avg. ↑ | $\Delta$ ↑ | sec/batch ↓ | |
| | | ARC-E | ARC-C | BoolQ | OBQA | PIQA | | | train | infer |
|---|---|---|---|---|---|---|---|---|---|---|
| *LLaMA-2 7B* | | | | | | | | | | |
| LoRA | 2.75% | 79.3 | 67.5 | 72.0 | 79.4 | 81.6 | 76.0 | - | 0.64 | |
| LoRAMoE | 2.83% | 79.0 | 65.0 | 71.8 | 79.8 | 82.6 | 75.7 | -0.3 | 1.62 | |
| MoLA | 2.83% | 77.9 | 65.0 | 71.0 | 76.8 | 83.9 | 74.9 | -1.1 | 1.47 | |
| Gated LoRA (top-32) | 2.75% | 82.6 | 66.6 | 72.2 | 78.4 | 83.8 | **76.7** | **+0.7** | 0.68 | |
| Gated LoRA (ReLU) | 2.75% | 81.1 | 68.2 | 71.0 | 79.4 | 83.9 | **76.7** | **+0.7** | 0.68 | |
| *LLaMA-2 13B* | | | | | | | | | | |
| LoRA | 2.26% | 83.2 | 71.9 | 75.2 | 84.6 | 85.1 | 80.0 | - | 0.85 | |
| LoRAMoE | 2.32% | 83.7 | 73.2 | 74.1 | 84.4 | 86.5 | 80.3 | +0.3 | 2.55 | |
| MoLA | 2.32% | 85.1 | 72.4 | 75.2 | 83.4 | 85.7 | 80.4 | +0.4 | 2.11 | |
| Gated LoRA (top-32) | 2.26% | 84.0 | 72.9 | 74.0 | 85.4 | 86.0 | **80.5** | **+0.5** | 1.16 | |
| Gated LoRA (ReLU) | 2.26% | 83.8 | 72.9 | 74.1 | 84.8 | 85.7 | 80.2 | +0.2 | 1.08 | |
| *LLaMA-3.1 8B* | | | | | | | | | | |
| LoRA | 2.82% | 84.9 | 75.2 | 73.8 | 83.8 | 86.8 | 80.9 | - | 0.61 | 1.22 |
| LoRAMoE | 2.89% | 87.0 | 77.1 | 74.9 | 84.4 | 89.1 | 82.5 | +1.6 | 1.60 | 3.09 |
| MoLA | 2.89% | 88.5 | 77.3 | 75.2 | 85.8 | 87.5 | 82.8 | +1.9 | 1.42 | 1.92 |
| Gated LoRA (top-32) | 2.82% | 87.0 | 79.7 | 75.0 | 87.2 | 88.7 | 83.5 | +2.6 | 0.64 | 1.33 |
| Gated LoRA (ReLU) | 2.82% | 87.7 | 79.9 | 74.9 | 87.6 | 89.0 | **83.8** | **+2.9** | 0.68 | 1.26 |

**Model.** Following prior works (Li et al., 2024; Dou et al., 2024), we adopt LLaMA-2 7B[2], LLaMA-2 13B[3], and LLaMA-3.1 8B[4] as our backbones (Touvron et al., 2023b; Meta, 2024). We employ Gated LoRA with 128-rank with two types of gating mechanisms: a **top-32** rank activation mechanism with orthogonal regularization with $\lambda_2 = 1.0$, and **ReLU** gating with sparsity regularization with $\lambda_1 = 1.0$. A target sparsity of $3/4$ is used to align with the top-32 activation over 128 total ranks. We applied the adapters exclusively to the FeedForward layers of LLaMA, while the attention layers remain untouched and frozen for all experiments.

**Training.** We use MoE-PEFT, an open-source framework developed by TUDB-Labs, for both training and evaluation.[5] For multi-task fine-tuning, we construct a training set by sampling an equal number of examples from ARC-E, ARC-C, BoolQ, OBQA, and PIQA, resulting in a combined dataset of 33,867 samples. For single-task fine-tuning, we utilize the full training split of each respective dataset. All models are trained for 2 epochs with a batch size of 16 and 2 gradient accumulation steps, for both single-task and multi-task settings. We used AdamW optimizer with a constant learning rate of $2e-4$. Detailed training settings are provided in Appendix A.2

## 5 EXPERIMENT RESULTS

We evaluate Gated LoRA across multi-task and single-task fine-tuning settings using three backbone models. Detailed comparisons with baseline methods and ablation studies are provided in the following subsections: LLaMA-2 7B/13B, and LLaMA-3.1 8B.

### 5.1 MULTI-TASK FINE-TUNING

Table 1 presents a comparison of PEFT methods on multi-task fine-tuning using three backbone models and five benchmark datasets. We compare our proposed Gated LoRA against standard LoRA, softmax-gated LoRAMoE, and top-32 gated MoLA. For all methods, the total number of ranks per adapter is fixed at 128 to ensure a fair comparison.

---

[2] https://huggingface.co/meta-llama/Llama-2-7b-hf

[3] https://huggingface.co/meta-llama/Llama-2-13b-hf

[4] https://huggingface.co/meta-llama/Llama-3.1-8B

[5] https://github.com/TUDB-Labs/MoE-PEFT

Table 2: Accuracy comparison of PEFT methods for **single-task fine-tuning** with LLaMA backbones. We use a total of 128 ranks for both LoRA and Gated LoRA. For Gated LoRA, we use two types: top-32 activation and dynamic ReLU gating with target sparsity of 0.75.

| Method | Accuracy (%) ↑ | | | | | | | | | avg. ↑ |
|---|---|---|---|---|---|---|---|---|---|---|
| | ARC-E | ARC-C | BoolQ | OBQA | PIQA | SIQA | HellaS | WinoG | CSQA | |
| *LLaMA-2 7B* | | | | | | | | | | |
| LoRA (Hu et al., 2022) | 77.0 | 53.1 | 72.1 | 76.6 | 82.2 | 78.1 | 92.8 | 50.0 | 74.7 | 73.0 |
| Gated LoRA (top-32) | 77.4 | 52.9 | 72.9 | 78.4 | 81.2 | 78.6 | 92.1 | 76.4 | 77.8 | 76.4 |
| Gated LoRA (ReLU) | 77.1 | 52.0 | 73.1 | 76.8 | 82.8 | 79.1 | 93.4 | 77.3 | 77.5 | **76.6** |
| *LLaMA-2 13B* | | | | | | | | | | |
| LoRA (Hu et al., 2022) | 80.9 | 66.6 | 74.6 | 82.6 | 84.9 | 79.5 | 94.1 | 82.4 | 80.2 | 80.6 |
| Gated LoRA (top-32) | 81.3 | 68.3 | 73.1 | 83.4 | 85.7 | 79.5 | 93.8 | 83.8 | 78.6 | 80.8 |
| Gated LoRA (ReLU) | 82.1 | 67.4 | 75.2 | 81.4 | 85.0 | 80.0 | 95.0 | 83.9 | 79.4 | **81.0** |
| *LLaMA-3.1 8B* | | | | | | | | | | |
| LoRA (Hu et al., 2022) | 87.2 | 78.2 | 74.4 | 85.4 | 88.0 | 78.7 | 95.0 | 83.0 | 80.1 | 83.3 |
| Gated LoRA (top-32) | 87.2 | 78.3 | 75.9 | 86.8 | 85.9 | 79.4 | 94.1 | 85.5 | 81.5 | 83.8 |
| Gated LoRA (ReLU) | 85.9 | 79.0 | 75.6 | 85.6 | 88.2 | 79.0 | 95.9 | 84.8 | 80.5 | **84.9** |

Across all backbones, Gated LoRA consistently achieves the highest or equal-best average accuracy. Notably, for LLaMA-3.1 8B, the ReLU-gated variant achieves an average accuracy of 83.8%, outperforming all baselines by up to +2.9%p. Even in the more constrained top-32 variant, Gated LoRA surpasses LoRAMoE and MoLA on all three backbones. These results highlight the effectiveness of input-dependent rank selection in improving generalization across diverse tasks.

**Efficiency Comparison** We report the trainable–parameter ratio, training time per iteration, and inference latency. Because the selector is computed from the same low-rank projection as LoRA, Gated LoRA adds no extra trainable parameters beyond LoRA. On LLaMA-3.1 8B, Gated LoRA adds only +0.03–0.07 sec/batch during training and +0.04–0.11 sec/batch at inference relative to LoRA; by contrast, LoRAMoE and MoLA add +0.99 and +0.81 sec/batch in training and +1.87 and +0.70 sec/batch in inference latency, respectively.

## 5.2 SINGLE-TASK FINE-TUNING

Table 2 reports the accuracy of various PEFT methods on single-task fine-tuning across nine downstream tasks and three different LLaMA backbones. We compare standard LoRA with two variants of our proposed Gated LoRA, each using a total of 128 ranks per adapter.

Across all backbones, Gated LoRA with ReLU gating achieves the highest average accuracy, improving over the standard LoRA baselines by +1.6-3.6%p depending on the backbone. The top-32 gated variant also matches or exceeds standard LoRA baselines, with an average accuracy improvement of up to +3.4%p. These results demonstrate that input-dependent gating can be beneficial not only in multi-task fine-tuning but even in single-task scenarios, likely by enabling better specialization across example types within each task. The consistent gains of ReLU gating further suggest that soft, dynamic sparsity may help capture fine-grained variations in reasoning patterns, even without explicit task-switching. Overall, Gated LoRA enhances fine-tuning performance while maintaining strong parameter efficiency.

## 6 ABLATION STUDY

### 6.1 SEPARATE GATING ABLATION

Table 3 and Table 4 compare our method against a model with a variant with a separate gating layer ($W_g \neq A$) on multi-task and single-task fine-tuning, respectively, with LLaMA-3.1 8B backbone. The gating layer $W_g$ is a linear map with the same shape as $A$, and both utilize top-32 activation.

The results show that our dual-purpose design ($W_g = A$) is not just more parameter-efficient but also more effective, improving average accuracy by +0.7%p (multi-task) and +0.6%p (single-task). We attribute this to a favorable inductive bias: tying the gating decision directly to the projection $Ax$

Table 3: Ablation on using a separate gating layer $W_g$ of Gated LoRA on **multi-task fine-tuning** with LLaMA-3.1 8B. All models use 128 total ranks with top-32 activation. **Params.%** is the proportion of trainable parameters. **avg.** denotes average accuracy.

| Method | Params. % | Accuracy (%) ↑ | | | | | avg. ↑ |
|---|---|---|---|---|---|---|---|
| | | ARC-E | ARC-C | BoolQ | OBQA | PIQA | |
| $W_g \neq A$ | 3.97 | 87.8 | 77.3 | 73.5 | 84.2 | 86.0 | 81.8 |
| $W_g = A$ | 2.82 | 86.8 | 77.4 | 74.6 | 86.8 | 86.9 | **82.5** |

Table 4: Ablation on using a separate gating layer $W_g$ on **single-task fine-tuning** with LLaMA-3.1 8B. All models use 128 total ranks with top-32 activation. **avg.** denotes average accuracy.

| Method | Accuracy (%) ↑ | | | | | | | | | avg. ↑ |
|---|---|---|---|---|---|---|---|---|---|---|
| | ARC-E | ARC-C | BoolQ | OBQA | PIQA | SIQA | HellaS | WinoG | CSQA | |
| $W_g \neq A$ | 87.0 | 76.9 | 74.9 | 84.6 | 87.3 | 77.7 | 94.4 | 85.3 | 80.3 | 83.2 |
| $W_g = A$ | 87.2 | 78.3 | 75.9 | 86.8 | 85.9 | 79.4 | 94.1 | 85.5 | 81.5 | **83.8** |

Table 5: Ablation study on gating mechanisms and regularization for multi-task fine-tuning with LLaMA-2 7B using Gated LoRA. **Gate** refers to the gating type (top-$k$ vs. ReLU), **Orth.** indicates whether orthogonal loss is applied, and **Spars.** indicates whether sparsity loss is applied.

| Gate | Regularization | | Accuracy (%) ↑ | | | | | avg. ↑ |
|---|---|---|---|---|---|---|---|---|
| | Orth. | Spars. | ARC-E | ARC-C | BoolQ | OBQA | PIQA | |
| top-32 | | | 82.4 | 67.2 | 71.2 | 77.6 | 84.1 | 76.5 |
| | ✓ | | 82.6 | 66.6 | 72.2 | 78.4 | 83.8 | **76.7** |
| | | ✓ | 80.8 | 67.7 | 71.5 | 77.0 | 83.0 | 76.0 |
| | ✓ | ✓ | 80.9 | 66.2 | 72.0 | 78.0 | 83.6 | 76.2 |
| ReLU | | | 79.3 | 63.8 | 72.5 | 76.6 | 83.4 | 75.1 |
| | ✓ | | 79.8 | 65.9 | 70.9 | 78.4 | 83.1 | 75.6 |
| | | ✓ | 81.2 | 68.2 | 71.0 | 79.4 | 83.9 | **76.7** |
| | ✓ | ✓ | 81.7 | 66.6 | 71.6 | 79.2 | 83.8 | 76.6 |

aligns the routing and update subspaces, encouraging the model to learn a more robust, unified representation and simplifying the overall optimization landscape. These results support the integrated, router-free architecture as a stronger alternative to externally routed variants.

## 6.2 ANALYSIS OF GATING AND REGULARIZATIONS

Table 5 presents an ablation study analyzing the impact of different gating mechanisms and regularization strategies in Gated LoRA, evaluated on multi-task fine-tuning with LLaMA-2 7B.

**Effect of Gating Mechanism.** Without any regularization, top-$k$ gating outperforms ReLU gating (76.5% vs. 75.1%), suggesting that explicit selection of the strongest rank components is more effective than ReLU's simple activation threshold, which can lead to overly dense activations. However, when paired with appropriate regularization, ReLU gating matches the performance of the best top-$k$ configuration, demonstrating that it can be highly effective once sparsity is controlled.

**Effect of Orthogonal Regularization.** Applying orthogonal regularization not only improves the average performance under both gating mechanisms (+0.2%p for top-$k$ gating, +0.5%p for ReLU gating), but also smooths per-task outcomes: in the top-$k$ setting, it reduces ARC-C from 67.2% to 66.6% and PIQA from 84.1% to 83.8%, seemingly trading off gains in dominant tasks to benefit others. These results suggest that orthogonal constraints reduce expert redundancy and promote broader generalization rather than overfitting to a subset of tasks.

**Effect of Sparsity Regularization.** For ReLU gating, adding sparsity regularization boosts performance, reaching the highest average accuracy (76.7%) among all ReLU configurations. This

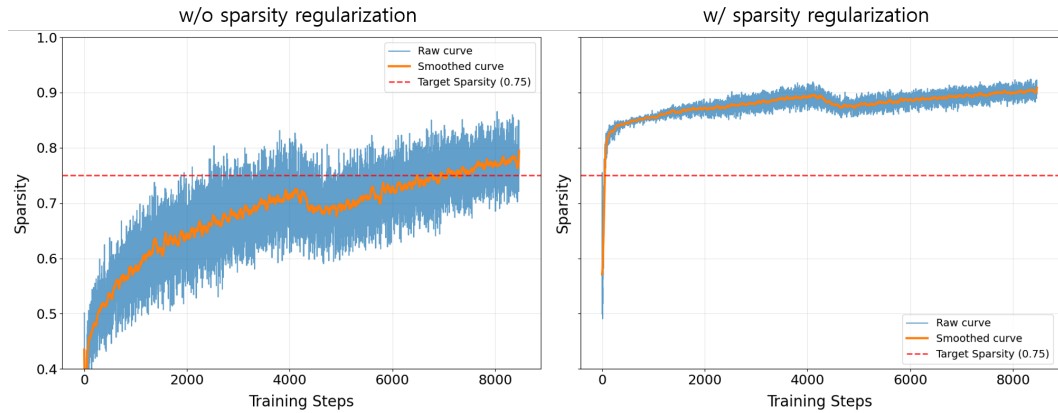

Figure 2: Activation sparsity of ReLU-gated Gated LoRA with (right) and without (left) sparsity regularization, measured over training iterations. The target sparsity for regularization is set to 0.75, indicated by the horizontal dotted red line. The blue curve represents the raw sparsity values at each iteration, while the orange curve shows the moving average for smoothing.

aligns with our expectation that sparsity regularization compensates for the lack of sparsity control, improving selection efficiency and enhancing performance. In top-$k$ gating, however, additional sparsity loss is redundant and can even be harmful (from 76.5% to 76.0%), since sparsity is already enforced by construction.

See Appendix D for additional ablations on $(r, k)$ configurations.

### 6.3 ANALYSIS OF ACTIVATION SPARSITY

To better understand the role of our sparsity regularization, we examine the activation dynamics of the ReLU selector during multi-task fine-tuning.

Figure 2 illustrates the evolution of activation sparsity in ReLU-gated LoRA with and without sparsity regularization over training in multi-task fine-tuning with LLaMA-3.1 8B. Sparsity is measured as the proportion of zero elements in $u(x) = \text{ReLU}(\boldsymbol{A}x)$. The red dashed line indicates the target sparsity level (0.75) used for regularization.

Interestingly, even without explicit regularization, ReLU gating tends to increase sparsity over time, with the average sparsity approaching the target value. This observation implies that sparse rank activation may be an emergent property of LLM optimization, with the training dynamics naturally favoring sparser expert usage even without explicit sparsity constraints. This may help explain why top-$k$ gating performs well in Gated LoRA, despite utilizing only a fixed subset of ranks at each step. However, as shown by the blue curve, the sparsity fluctuates significantly across training steps, indicating instability in expert activation patterns.

In contrast, applying $\ell_1$ regularization not only encourages higher sparsity, reaching close to 0.9 on average, but also stabilizes the activation pattern throughout training. The reduced variance suggests that sparsity regularization acts as a form of regularization for the gating dynamics, helping Gated LoRA converge to more consistent and efficient rank selection.

See Appendix G for additional analysis of orthogonality and (activation) sparsity.

## 7 CONCLUSION

We presented Gated LoRA, a lightweight extension of LoRA that performs input-adaptive, rank-wise updates by reusing the down-projection for both feature extraction and gate production. Interpreting each rank-1 factor as a fine-grained feature direction enables conditional computation, without additional routing parameters and with negligible latency overhead. Across nine multi- and single-task benchmarks, Gated LoRA matches or surpasses standard LoRA and MoE-based PEFT

baselines under the same compute budget. Ablations isolate the benefit of rank-wise gating and show that light sparsity and orthogonality regularization stabilize training and yield more balanced cross-task performance. These results underscore dynamic, rank-wise adaptation as a promising direction for improving the robustness and efficiency of parameter-efficient LLM fine-tuning.

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

## ETHICS STATEMENT

We followed community norms for reproducibility and research integrity consistent with the ICLR Code of Ethics. We conducted the research with academic integrity and appropriate citation practices; no plagiarism or misappropriation of others' work occurred. We used only publicly available NLP benchmarks, did not collect new human-subject data, and did not process personally identifiable information. To the best of our knowledge, all datasets and any third-party assets used comply with their licenses; we cite sources and versions in the appendix.

## THE USE OF LLMS

Large Language Models were exculsively to correct grammar and refine the wording of the manuscript text. No original ideas, analyses, or passages were generated by these tools. All authors reviewed AI-assisted edits and accept full responsibility for the final manuscript.

## A    EXPERIMENT DETAILS

### A.1    DATASETS

Table 6 provides an overview of the nine benchmark datasets used in our experiments, including task type, domain, and the number of training and test samples. All datasets are publicly available and can be downloaded from the Hugging Face Hub.

Table 6: Dataset Details.

| Benchmark | Task | Domain | # Train | # Test |
|---|---|---|---|---|
| ARC-E (Clark et al., 2018) | Question answering | Elementary science | 2,250 | 2,380 |
| ARC-C (Clark et al., 2018) | Question answering | Elementary science | 1,120 | 1,170 |
| BoolQ (Clark et al., 2019) | Question answering | Wikipedia | 9,427 | 3,270 |
| OpenBookQA (Mihaylov et al., 2018) | Question answering | Elementary science | 4,957 | 500 |
| PIQA (Bisk et al., 2019) | Question answering | Physical Commonsense | 16,100 | 1,840 |
| SIQA (Sap et al., 2019) | Question answering | Social Commonsense | 33,410 | 1,954 |
| HellaSwag (Zellers et al., 2019) | Sentence completion | WikiHow and video captions | 39,905 | 10,042 |
| WinoGrande (Sakaguchi et al., 2021) | Fill-in-the-Blank | Pronoun resolution problems | 9,248 | 1,267 |
| CommonSenseQA (Talmor et al., 2019) | Question answering | Commonsense | 9,741 | 1,140 |

### A.2    HYPERPARAMETERS

Table 7 presents the hyperparameter configurations used for all baseline methods and Gated LoRA with both gating variants.[6] For additional details on the implementation, we refer readers to the MoE-PEFT framework developed by TUDB-Labs.

## B    DISCUSSION

### B.1    RELATION TO PRIOR WORKS

SMoRA (Zhao et al., 2025b) is a recently proposed method that, like our approach, treats each rank-1 component in LoRA as an individual expert and applies separate top-$k$ router to activate ranks based on the input dynamically. While the high-level motivation appears similar, **Gated LoRA was developed independently with a distinct focus on simplicity and integration**, resulting in notable differences in design and implementation.

The key difference lies in the routing mechanism. SMoRA, like any other MoE frameworks, introduces a separate routing network with additional parameters, including a load-balancing router

---

[6]Note that the 'gate' in target modules is a linear layer for SwiGLU activation inside the LLaMA FFN module, and not a gating module used in MoE or Gated LoRA.

Table 7: Hyperparameter Configurations.

| Hyperparameter | LoRA | LoRAMoE | MoLA | Gated LoRA (top-$k$) | Gated LoRA (ReLU) |
|---|---|---|---|---|---|
| cutoff length | | | | 512 | |
| optimizer | | | | AdamW | |
| learning rate | | | | 2e-4 | |
| scheduler | | | | constant | |
| batch size | | | | 16 | |
| accumulation steps | | | | 2 | |
| epochs | | | | 2 | |
| LoRA dropout | | | | 0.05 | |
| target modules | | | | gate, up, down | |
| LoRA rank | 128 | 16 | 16 | 128 | 128 |
| LoRA experts | - | 8 | 8 | - | - |
| top-$k$ | - | - | 2 | 32 | - |

Table 8: Accuracy comparison of rank-wise routed LoRA on multi-task fine-tuning using three different backbones. All models use a total of 128 ranks. **avg.** denotes the average accuracy.

| Method | Accuracy (%) ↑ | | | | | avg. ↑ |
|---|---|---|---|---|---|---|
| | **ARC-E** | **ARC-C** | **BoolQ** | **OBQA** | **PIQA** | |
| *LLaMA-2 7B* | | | | | | |
| SMoRA (top-32) (Zhao et al., 2025b) | 39.9 | 32.4 | 61.2 | 33.8 | 57.0 | 44.9 |
| Gated LoRA (top-32) | 82.6 | 66.6 | 72.2 | 78.4 | 83.8 | 76.7 |
| Gated LoRA (ReLU) | 81.1 | 68.2 | 71.0 | 79.4 | 83.9 | 76.7 |
| *LLaMA-2 13B* | | | | | | |
| SMoRA (top-32) (Zhao et al., 2025b) | 67.4 | 51.8 | 61.6 | 41.8 | 66.2 | 57.8 |
| Gated LoRA (top-32) | 84.0 | 72.9 | 74.0 | 85.4 | 86.0 | 80.5 |
| Gated LoRA (ReLU) | 83.8 | 72.9 | 74.1 | 84.8 | 85.7 | 80.2 |
| *LLaMA-3.1 8B* | | | | | | |
| SMoRA (top-32) (Zhao et al., 2025b) | 81.5 | 67.9 | 57.2 | 68.8 | 66.6 | 68.4 |
| Gated LoRA (top-32) | 86.8 | 78.7 | 74.6 | 87.2 | 88.6 | 83.2 |
| Gated LoRA (ReLU) | 87.7 | 79.9 | 74.9 | 87.6 | 89.0 | 83.8 |

bias (Wang et al., 2024). In contrast, Gated LoRA adopts a *dual-purpose design* in which the same low-rank projection matrix $A$ is reused for both adaptation and selection. This eliminates the need for external routers or auxiliary objectives, yielding a more lightweight and unified architecture.

By reinterpreting LoRA's rank components as fine-grained feature directions, Gated LoRA enables input-dependent adaptation through a streamlined mechanism that requires no architectural modifications beyond standard LoRA. This design highlights a novel and conceptually grounded direction within the broader space of parameter-efficient sparse expert models. The results from the separate gating ablations (Table 3 and Table 4) further justify our dual-purpose design and highlight the differences from prior work.

At the time of our experiments, SMoRA had not released public code, and we were unable to faithfully reproduce its reported results. As such, we chose not to include SMoRA in our main empirical comparisons, as doing so would have been unfair to both our work and SMoRA. Nonetheless, for completeness, we report the results of our own naïve re-implementation of SMoRA using a total of 128 ranks with top-32 routing on three backbones in Table 8. We believe future comparisons will be valuable once official implementations and reproducible benchmarks are available.

## C  INTUITIVE RATIONALE OF DESIGN CHOICES

While not a formal proof, this section provides intuition for two key components of Gated LoRA: the top-$k$ gating strategy and the orthogonality constraint on the projection matrix $\boldsymbol{B}$.

### C.1  TOP-K GATING MAXIMIZES UPDATE NORM

Let $x \in \mathbb{R}^{d_{\text{in}}}$ be the input feature. Gated LoRA defines a sparse low-rank update of the form:

$$\Delta h(x, m) = \boldsymbol{B} \cdot \text{diag}(m) \cdot \boldsymbol{A}x, \tag{10}$$

where $\boldsymbol{A} \in \mathbb{R}^{r \times d_{\text{in}}}$ and $\boldsymbol{B} \in \mathbb{R}^{d_{\text{out}} \times r}$ are LoRA matrices and $m \in \{0, 1\}^r$ is a binary gating vector with sparsity constraint $\|m\|_0 = k$. We aim to choose the top-$k$ directions that yield the most impactful update by maximizing the squared norm:

$$\|\Delta h(x, m)\|_2^2 = \left\| \sum_{i=1}^{r} m_i b_i (a_i^\top x) \right\|_2^2, \tag{11}$$

where $b_i$ and $a_i^\top$ denote the $i$-th column of $\boldsymbol{B}$ and row of $\boldsymbol{A}$, respectively.

**Observation.**  If the columns $\{b_i\}_{i=1}^r$ of $\boldsymbol{B}$ are orthonormal, then the top-$k$ gating vector $m$ that maximizes $\|\Delta h(x, m)\|_2^2$ selects the $k$ indices with the largest squared activation $(a_i^\top x)^2$.

**Intuition.**  Under orthonormality, the cross-terms containing $b_i^\top b_j (i \neq j)$ vanish:

$$\|\Delta h(x, m)\|_2^2 = \sum_{i=1}^{r} m_i (a_i^\top x)^2. \tag{12}$$

Thus, maximizing the update norm reduces to selecting the $k$ largest values of $(a_i^\top x)^2$, which corresponds precisely to top-$k$ gating based on the magnitude of projection activations.

In the general (i.e., non-orthogonal) case, the squared norm includes the cross terms:

$$\|\Delta h(x, m)\|_2^2 = \sum_{i=1}^{r} m_i (a_i^\top x)^2 \|b_i\|_2^2 + \sum_{i \neq j} m_i m_j (a_i^\top x)(a_j^\top x)(b_i^\top b_j), \tag{13}$$

where the second term introduces interference. In this case, the top-$k$ strategy maximizes a diagonal surrogate objective:

$$\sum_{i=1}^{r} m_i^2 (a_i^\top x)^2 \|b_i\|_2^2, \tag{14}$$

which approximates the true norm well when the off-diagonal terms $b_i^\top b_j$ are small.

Although we cannot guarantee optimality when $\boldsymbol{B}$ is not orthogonal, empirical results in Section 5 suggest that this surrogate remains well-correlated with performance. Thus, top-$k$ gating serves as a practical and effective heuristic, even when $\boldsymbol{B}$ is only approximately orthogonal.

### C.2  ORTHOGONALITY REDUCES RANK-WISE INTERFERENCE

To reduce interference between ranks, we encourage orthogonality among the columns of $B$. This can be formalized by analyzing the expected interaction between directions under a task-specific input distribution $p(t)$. We define the expected interference energy between directions $i$ and $j$ as:

$$\mathcal{E}_{i,j} = \mathbb{E}_{x \sim p_t}[(a_i^\top x)(a_j^\top x)(b_i^\top b_j)]. \tag{15}$$

Applying the Cauchy-Schwarz inequality yields:

$$|\mathcal{E}_{i,j}| \leq \|b_i\|_2 \|b_j\|_2 \, \mathbb{E}_{x \sim p_t} \left[ |a_i^\top x| \cdot |a_j^\top x| \right], \tag{16}$$

Table 9: Ablation study on configurations of Gated LoRA on multi-task fine-tuning with LLaMA-3.1 8B with **top-$k$ gating**. $r$ refers to the total number of ranks, $k$ refers to the number of active ranks.

| $r$ | $k$ | Accuracy (%) ↑ | | | | | avg. ↑ |
|---|---|---|---|---|---|---|---|
| | | ARC-E | ARC-C | BoolQ | OBQA | PIQA | |
| 128 | 8 | 85.7 | 76.5 | 71.8 | 84.8 | 87.4 | 81.2 |
| | 16 | 85.9 | 75.1 | 72.0 | 86.8 | 86.5 | 81.3 |
| | 32 | 86.8 | 77.4 | 74.6 | 86.8 | 86.9 | **82.5** |
| | 64 | 83.5 | 74.8 | 70.6 | 86.2 | 85.9 | 80.2 |
| | 128 | 84.9 | 75.2 | 73.8 | 83.8 | 86.8 | 80.9 |
| 64 | 8 | 84.8 | 74.8 | 74.1 | 85.0 | 88.0 | 80.5 |
| | 16 | 85.5 | 77.2 | 75.1 | 82.6 | 88.2 | **80.8** |
| | 32 | 85.6 | 75.6 | 72.4 | 84.4 | 87.3 | 79.9 |
| | 64 | 84.6 | 74.7 | 72.3 | 85.0 | 86.7 | 79.7 |
| 32 | 8 | 88.3 | 78.8 | 74.8 | 86.2 | 87.6 | **83.1** |
| | 16 | 86.1 | 78.2 | 74.5 | 86.4 | 87.4 | 82.5 |
| | 32 | 84.0 | 74.5 | 71.5 | 85.8 | 85.9 | 80.3 |

showing that interference grows with the cosine similarity between $b_i$ and $b_j$. Reducing $b_i^\top b_j$ for $i \neq j$ tightens this upper bound.

To promote orthogonality, we apply the following regularization term:

$$\Omega(B) = \lambda \sum_{i \neq j} (b_i^\top b_j)^2. \tag{17}$$

This regularization explicitly discourages correlated directions among gated components, thereby reducing within-task interference and encouraging expert specialization.

## D  EFFECT OF THE TOTAL RANK AND ACTIVE RANKS

We conduct additional experiments with a much lighter budget setting, with various numbers of active ranks. Table 9 and Table 10 present ablation studies analyzing the impact of different total rank $r$ and the number of active ranks $k$ in Gated LoRA with top-$k$ gating and ReLU gating, respectively, evaluated on multi-task fine-tuning with LLaMA-3.1 8B.

Across nearly all configurations, using a sparse subset of ranks ($k < r$) outperforms using the full rank, confirming that dynamic gating is a powerful tool for improving performance. Interestingly, the two gating mechanisms show different sensitivities to the total rank $r$. For top-$k$ gating, the best performance (83.1) is achieved with the smallest total rank of r=32. The optimal sparsity ratio is also remarkably consistent at around 25% (e.g., k=32 for r=128, k=16 for r=64, and k=8 for r=32). This suggests that for top-k gating, a smaller total rank acts as a strong and beneficial regularizer. Conversely, for ReLU gating, the overall best performance (83.8) is achieved with the largest total rank of r=128. The flexible, dynamic nature of ReLU gating appears to be a powerful regularizer in its own right, allowing the model to effectively leverage a larger pool of rank directions without overfitting.

## E  FINE-TUNIING ON PHI3 AND QWEN2

To assess whether the benefits of Gated LoRA extend beyond the LLaMA family, we additionally evaluate two backbones: Phi-3-Mini-4K-Instruct[7] and Qwen2-7B[8]. Table 11 reports multi-task fine-tuning results under the same training protocol as our main experiments, including the same datasets and total rank budget (128 ranks per adapter).

---

[7] https://huggingface.co/microsoft/Phi-3-mini-4k-instruct

[8] https://huggingface.co/Qwen/Qwen2-7B

Table 10: Ablation study on configurations of Gated LoRA on multi-task fine-tuning with LLaMA-3.1 8B with **ReLU gating** with sparsity regularization. $r$ refers to the total number of ranks, $k$ refers to the target number of active ranks.

| $r$ | $k$ | Accuracy (%) ↑ | | | | | avg. ↑ |
|---|---|---|---|---|---|---|---|
| | | ARC-E | ARC-C | BoolQ | OBQA | PIQA | |
| 128 | 8 | 85.8 | 78.8 | 74.7 | 86.4 | 86.9 | 82.5 |
| | 16 | 87.1 | 79.5 | 74.9 | 85.6 | 88.1 | 83.0 |
| | 32 | 87.7 | 79.9 | 74.9 | 87.6 | 89.0 | **83.8** |
| | 64 | 86.7 | 78.5 | 75.0 | 85.6 | 88.3 | 82.8 |
| | 128 | 84.9 | 75.2 | 73.8 | 83.8 | 86.8 | 80.9 |
| 64 | 8 | 86.9 | 80.2 | 75.1 | 86.4 | 88.4 | 82.5 |
| | 16 | 86.5 | 80.6 | 74.6 | 87.6 | 88.8 | **82.9** |
| | 32 | 88.3 | 79.4 | 75.5 | 86.8 | 88.6 | 82.6 |
| | 64 | 84.6 | 74.7 | 72.3 | 85.0 | 86.7 | 79.7 |
| 32 | 8 | 86.9 | 78.4 | 72.7 | 86.4 | 87.4 | 82.4 |
| | 16 | 87.2 | 79.5 | 74.6 | 87.6 | 87.9 | **83.4** |
| | 32 | 84.0 | 74.5 | 71.5 | 85.8 | 85.9 | 80.3 |

Table 11: Accuracy on **multi-task** fine-tuning with diverse backbones. All models use 128 total ranks. $\Delta$ is the average accuracy improvement over the LoRA baseline (percentage points).

| Method | Accuracy (%) ↑ | | | | | avg. ↑ | $\Delta$ ↑ |
|---|---|---|---|---|---|---|---|
| | ARC-E | ARC-C | BoolQ | OBQA | PIQA | | |
| *Phi-3-Mini-4K-Instruct* | | | | | | | |
| LoRA | 90.1 | 82.5 | 67.9 | 86.8 | 86.2 | 82.7 | - |
| LoRAMoE | 90.8 | 84.0 | 69.8 | 87.6 | 86.3 | 83.7 | +1.0 |
| MoLA | 90.2 | 84.2 | 70.0 | 86.4 | 86.1 | 83.4 | +0.7 |
| Gated LoRA (top-32) | 93.4 | 84.7 | 69.8 | 86.6 | 86.3 | **84.2** | **+1.5** |
| Gated LoRA (ReLU) | 91.8 | 83.9 | 69.1 | 87.4 | 86.2 | **83.7** | +1.0 |
| *Qwen2-7B* | | | | | | | |
| LoRA | 87.9 | 78.8 | 71.6 | 88.4 | 88.4 | 83.0 | - |
| LoRAMoE | 89.6 | 84.4 | 73.4 | 90.8 | 89.8 | 85.6 | +2.6 |
| MoLA | 89.7 | 82.7 | 74.0 | 88.8 | 88.5 | 84.8 | +1.7 |
| Gated LoRA (top-32) | 90.2 | 86.1 | 73.2 | 91.6 | 90.1 | **86.2** | **+3.2** |
| Gated LoRA (ReLU) | 90.3 | 86.0 | 72.8 | 91.4 | 89.3 | 86.0 | +3.0 |

Across both backbones, Gated LoRA attains the highest or equal-best average accuracy among all methods under the same parameter budget. For Phi-3-Mini-4K-Instruct, the top-32 variant of Gated LoRA achieves an average accuracy of 84.2%, improving over the LoRA baseline by +1.5&-points and surpassing the best MoE-based baseline by +0.5%-points. For Qwen2-7B, while all MoE-based methods improve over the LoRA baseline, Gated LoRA (top-32) achieves the best overall performance with an average accuracy of 86.2%, corresponding to a +3.2%-point gain over LoRA and a +0.6%-point gain over LoRAMoE. These results confirm that rank-wise input-dependent gating remains effective across substantially different backbone architectures.

## F  FINE-TUNING ON MATHEMATICS BENCHMARK

To analyze the performance of our method on tasks requiring complex compositional reasoning, we expanded our evaluation to include GSM8K, a benchmark of high-school math problems. Table 12 and Table 13 present comparisons on multi-task and single-task fine-tuning, respectively, using the LLaMA-3.1 8B backbone. The experimental setup is identical to our main experiments, and we note that no chain-of-thought or few-shot prompting was used.

Table 12: Accuracy comparison of PEFT methods on **multi-task** fine-tuning, including GSM8K with LLaMA-3.1 8B. All models use 128 total ranks. **avg.** denotes average accuracy.

| Method | Accuracy (%) ↑ | | | | | | avg. ↑ |
|--------|------|------|------|------|------|-------|--------|
| | **ARC-E** | **ARC-C** | **BoolQ** | **OBQA** | **PIQA** | **GSM8K** | |
| LoRA (Hu et al., 2022) | 83.8 | 75.2 | 71.5 | 86.0 | 85.7 | 48.7 | 75.2 |
| LoRAMoE (Dou et al., 2024) | 84.7 | 76.3 | 74.7 | 83.6 | 87.3 | 50.1 | 76.1 |
| MoLA (Gao et al., 2024) | 84.0 | 74.6 | 74.8 | 83.8 | 87.2 | 49.4 | 75.6 |
| Gated LoRA (top-32) | 83.9 | 76.0 | 72.3 | 84.0 | 85.5 | 46.3 | 74.7 |
| Gated LoRA (ReLU) | 87.1 | 78.4 | 73.6 | 85.2 | 86.2 | 51.7 | **77.0** |

Table 13: Accuracy comparison of PEFT methods on GSM8K **single-task** fine-tuning with LLaMA-3.1 8B. All models use 128 total ranks. **avg.** denotes average accuracy.

| Method | Accuracy (%) ↑ |
|--------|------|
| | **GSM8K** |
| LoRA (Hu et al., 2022) | 48.4 |
| Gated LoRA (top-32) | 47.2 |
| Gated LoRA (ReLU) | 50.6 |

Table 12 and Table 13 present comparisons of PEFT methods on multi-task fine-tuning and single-task fine-tuning, respectively, using the LLaMA-3.1 8B backbone model, including the GSM8K benchmark. Experiment details are identical to the main experiments. Note that due to resource limitations, we did not use chain-of-thought prompting nor few-shot example prompting.

Across both multi-task and single-task settings, our Gated LoRA with ReLU dynamic gating consistently outperformed all baselines. In contrast, Gated LoRA with top-$k$ activation underperformed on this specific task. We hypothesize this is due to the rigid nature of top-$k$ selection being ill-suited for the compositional reasoning required in mathematical problem solving. The flexible and dynamic activation pattern of ReLU gating, which allows the model to recruit a variable number of experts per problem, proves to be a far more effective strategy when the task is challenging.

## G ORTHOGONALITY AND SPARSITY ANALYSIS

To further analyze the efficacy of our method, we measure two metrics: orthogonal deviation and activation sparsity. **Orthogonal Deviation** measures the non-orthogonality of a matrix $B$, which is a Frobenius norm of the difference between $B^\top B$ and the identity matrix,

$$\mathbf{OD}(B) = \|B^\top B - I\|_F. \tag{18}$$

For any input $x_i$ and any adapted layer $l$, we get a binary gating vector $g_{i,l} \in \{0,1\}^r$, which represents the set of active ranks. **Activation Sparsity** measures the ratio of the number of zero elements of the gating vector:

$$\mathbf{AS} = \mathbb{E}_{x_i \in \mathcal{D}, l \in \mathcal{L}, t \in \mathcal{T}} \left[ 1 - \frac{1}{r} \sum_{k=1}^{r} g_{i,l}^{(k)} \right], \tag{19}$$

averaged over test examples $\mathcal{D}$, layers $\mathcal{L}$, and tasks $\mathcal{T}$.

**Activation Overlap** is the average per-layer overlap, which is the mean Jaccard index over pairs of gating vectors of that layer:

$$\mathbf{AO} = \frac{1}{|\mathcal{L}||\mathcal{T}|} \sum_{l \in \mathcal{L}, t \in \mathcal{T}} \mathbb{E}_{x_i, x_j \sim \mathcal{D}_t, i \neq j} \left[ \frac{\sum_{k=1}^{r} g_{i,l}^{(k)} \cdot g_{j,l}^{(k)}}{\sum_{k=1}^{r} \max(g_{i,l}^{(k)}, g_{j,l}^{(k)})} \right]. \tag{20}$$

To further compare with random activation, we report the **normalized Activation Overlap**. For a random selector with empirical active ratio $\hat{\rho} = 1 - \mathbf{AS}$, the expected Jaccard for random vector

Table 14: Orthogonality and Sparsity measures for different gating and regularization configurations of Gated LoRA with total rank of 32 and 8 target active ranks. **Orth.** indicates whether orthogonal loss is applied, and **Spars.** indicates whether sparsity loss is applied. **OD** indicates orthogonal deviation, **AS** indicates activation sparsity, and **AO** indicates activation overlap. **nAO** indicates normalized activation overlap with the empirical sparsity.

| Gate | Regularization | | OD | AS | AO | nAO |
|------|------|------|------|------|------|------|
| | **Orth.** | **Spars.** | | | | |
| top-32 | | | 16.5 | 0.75 | 0.37 | 0.27 |
| | ✓ | | 14.5 | 0.75 | 0.36 | 0.25 |
| ReLU | | | 24.1 | 0.68 | 0.47 | 0.35 |
| | | ✓ | 19.8 | 0.90 | 0.39 | 0.36 |

pair is $J_{\text{rand}} \approx \hat{\rho}/(2 - \hat{\rho})$. The normalized score is computed as:

$$\mathbf{nAO} = \frac{\mathbf{AO} - J_{\text{rand}}}{1 - J_{\text{rand}}}, \tag{21}$$

so that 0 indicates random reuse and 1 indicates consistent gating.

Table 14 reports the orthogonal deviation, activation sparsity, and activation overlap of Gated LoRA on multi-task fine-tuning with the LLaMA-3.1 8B backbone model with different gating mechanisms and regularization configurations. We use a total rank of 32 with 8 active ranks to examine the orthogonality more thoroughly, while other training configurations are identical to the main experiment. All metrics are measured on test sets.

For top-$k$ gating, applying orthogonal loss successfully reduces the orthogonal deviation from 16.5 to 14.5, confirming that the regularization term is effective at making the feature directions in matrix $B$ more orthogonal to one another. For ReLU gating, applying the sparsity loss significantly increases the activation sparsity from 0.68 to 0.90, exceeding the target sparsity of 0.75. Interestingly, applying the sparsity loss also significantly reduces the orthogonal deviation from 24.1 to 19.8. This suggests the sparsity loss has a beneficial side-effect: forcing the model to use a sparser set of blueranks may indirectly encourage those active ranks to become more distinct and less redundant, explaining the performance degradation when applying sparsity loss to top-$k$ gated LoRA.

Activation overlap (Jaccard over active ranks) is consistent at around 0.36-0.47 across configurations. The normalized AO also shows consistent positive values, indicating the consistent reuse of ranks across examples, while still changing a non-trivial fraction per input. This provides strong evidence that Gated LoRA exhibits balanced selectivity, rather than collapsing into a fixed "superstar" subset. This supports our premise that fine-grained, rank-wise gating captures input variation better than coarse, block-wise activation.

