# OpenReview forum: "Gated LoRA: Dual-Purpose Projections for Parameter-Efficient Mini-Expert Fine-Tuning"
_ICLR.cc/2026/Conference — Submitted to ICLR 2026_

### Official Review · Reviewer_kaxG · 2025-10-18

**Soundness:** 2
**Presentation:** 3
**Contribution:** 1
**Rating:** 2
**Confidence:** 4

**Summary:**

The paper addresses the challenges in fine-tuning large language models (LLMs) where full fine-tuning is computationally expensive, leading to the adoption of parameter-efficient fine-tuning (PEFT) methods like Low-Rank Adaptation (LoRA). The proposed solution, Gated LoRA, treats each rank-1 component as a mini-expert and uses dual-purpose projections for both feature computation and rank selection, incorporating top-k or ReLU-based gating with sparsity and orthogonality regularization to enable fine-grained, adaptive updates.

**Strengths:**

1. The dual-purpose design reuses the down-projection matrix for both adaptation and gating, eliminating the need for separate routing parameters. This approach maintains parameter efficiency equivalent to standard LoRA while enabling input-dependent selectivity. It simplifies integration into existing frameworks without increasing training complexity.

2. Empirical results show consistent accuracy improvements over baselines like LoRA, LoRAMoE, and MoLA across multiple backbones and tasks. Gains reach up to 2.9 percentage points in multi-task settings and 3.6 in single-task fine-tuning. These enhancements demonstrate robustness in handling task diversity.


3. Negligible latency overhead during training and inference makes the method practical for real-world deployment. Compared to MoE-based alternatives, it adds minimal computational cost. This efficiency supports scalable application in resource-constrained environments.

**Weaknesses:**

1. Experiments use only LLaMA family backbones, potentially overlooking generalization to other architectures like GPT or BERT. Differences in model designs might affect gating efficacy. Testing on diverse LLMs would provide stronger evidence.

2. The paper samples equal examples for multi-task training, resulting in a modest dataset size. This may not capture real-world imbalances across tasks. Larger or varied sampling could better simulate practical scenarios.


3. Comparisons exclude some recent PEFT variants beyond the listed baselines. Emerging methods might offer competitive alternatives. Including them could highlight relative advantages more comprehensively.

4. The method underperforms on math benchmarks without chain-of-thought prompting. This suggests limitations in compositional reasoning tasks. Adapting gating for advanced prompting techniques might be necessary.

5. There have been some works considering regarding LoRA Adapters as experts to be routed [1,2]. However, these works are not discussed or compared.


[1] Sub-MoE: Efficient Mixture-of-Expert LLMs Compression via Subspace Expert Merging. Arxiv 2025.
[2] SLoRA: Scalable Serving of Thousands of LoRA Adapters. In MLsys 2024.

**Questions:**

See weaknesses.

---

> ### Author Response · Authors · 2025-12-01
>
> # Experiment Scope
>
> The reviewer notes that the original experiments used only LLaMA backbones and suggests that this may limit generalization to GPT/BERT-like architectures. In the revised manuscript, we have added multi-task experiments on two additional decoder-only backbones, **Phi-3** and **Qwen-2**. The results are summarized in **Table A:**
>
> **Table A:** Accuracy on multi-task fine-tuning with diverse backbones.
> | Backbone | Method | avg. (%) | gain (%p) |
> | --------------| ---------- | ------ | ------ |
> | Phi-3-Mini-4K-Instruct | LoRA | 82.7 | - |
> | | LoRAMoE | 83.7 | +1.0 |
> | | MoLA | 83.4 | +0.7 |
> | | Gated LoRA (top-32) | **84.2** | **+1.5** |
> | | Gated LoRA (ReLU) | 83.7 | +1.0 |
> | |
> | Qwen2-7B | LoRA | 83.0 | - |
> | | LoRAMoE | 85.6 | +2.6 |
> | | MoLA | 84.8 | +1.8 |
> | | Gated LoRA (top-32) | **86.2** | **+3.2** |
> | | Gated LoRA (ReLU) | 86.0 | +3.0 |
>
> Under the same five-task multi-task setting, Gated LoRA consistently improves over standard LoRA and also outperforms MoE-PEFT baselines on both Phi-3 and Qwen2. These additions show that Gated LoRA is **not specific to one model family** and can be applied to diverse decoder-only LLM architectures. Exploring encoder-only backbones (e.g., BERT-like models) is an interesting direction, but it goes beyond the PEFT scenario we focus on in this work.
>
>
> # GSM8K "Underperformance"
>
> The reviewer states that "the method underperforms on math benchmarks without chain-of-thought prompting." It is true that our absolute GSM8K scores (without CoT) are below those of fully fine-tuned or CoT-optimized systems reported in the literature, since designing specialized reasoning prompts is outside our scope.
>
> However, in **Table 12** and **Table 13**, within our PEFT baselines on LLaMA-3.1-8B, the **ReLU-gated** variant of Gated LoRA achieves the **best GSM8K accuracy** in both the multi-task and single-task settings, outperforming standard LoRA, LoRAMoE, and MoLA. The top-$k$ variant is weaker on GSM8K; we discuss that continuous gating appears better suited for compositional reasoning in our setup.
>
> Our aim is not to claim state-of-the-art GSM8K performance, but to show that **under identical prompting and PEFT budgets**, Gated LoRA can match or exceed MoE-PEFT baselines even on a math benchmark.
>
> # Balanced Data Sampling
>
> The reviewer points out that we use equal sampling of each dataset in multi-task training and that this may not reflect real-world imbalanced regimes. Our choice of equal sampling was intentional: it allows us to isolate the effect of gating while keeping the training budget comparable across backbones and baselines. We agree that heavily imbalanced settings could provide additional insights. However, implementing fully imbalanced regimes within the MoE-PEFT codebase we built upon was non-trivial and beyond the rebuttal timeframe. We leave a more detailed study of imbalanced multi-task training to future work.
>
> # Missing LoRA-as-expert Works
>
> The reviewer cites Sub-MoE [1] and SLoRA [2] as "works considering LoRA Adapters as experts to be routed". We appreciate these pointers, but we view them as addressing a different problem from the one studied in our paper.
> - **S-LoRA** is a **serving system** for deploying many task-specific LoRA adapters efficiently. Its focus is on unified paging, batching, and serving throughput/latency; it does not propose rank-wise token-level routing or a new PEFT architecture.
> - **Sub-MoE** is a **Mixture-of-Experts compression** method: it clusters and merges full FFN experts in a shared subspace to reduce the number of experts in an MoE backbone. It does not involve LoRA adapters and is not a PEFT method.
>
> By contrast, Gated LoRA is neither a serving system nor a MoE-compression technique, and does not target MoE backbones. Given this clear difference in scope and objectives, we do not consider Sub-MoE or SLoRA to be natural baselines or central prior art for our specific problem setting, so we chose not to expand the paper to cover them in detail.
>
> [1] Sub-MoE: Efficient Mixture-of-Expert LLMs Compression via Subspace Expert Merging. Arxiv 2025.
>
> [2] SLoRA: Scalable Serving of Thousands of LoRA Adapters. MLsys 2024.

---

### Official Review · Reviewer_j89t · 2025-10-31

**Soundness:** 3
**Presentation:** 3
**Contribution:** 2
**Rating:** 4
**Confidence:** 4

**Summary:**

The paper proposes a PEFT method named Gated LoRA. Its core idea is to reuse the standard LoRA down-projection matrix $A$ as a gating mechanism to dynamically select which rank-1 components (which the authors call "mini-experts") to activate. The selling point of this design is that it achieves input-adaptive sparse activation without introducing any additional trainable parameters. The authors demonstrate through experiments that this method achieves performance improvements over standard LoRA and other MoE-PEFT methods on several single-task and multi-task benchmarks.

**Strengths:**

1.	Design Simplicity: The "dual-purpose projection" design is very simple, achieving dynamic gating with zero parameter overhead, which is attractive from an engineering perspective.

2.	Parameter Efficiency: In the current research climate focused on PEFT parameter budgets, "not adding any trainable parameters" is a very strong selling point.

3.	Empirical Results: It must be acknowledged that the paper's experimental results are positive. It consistently outperforms standard LoRA across multiple benchmarks (though the improvement margin is around 2-3 points, not disruptive), which proves the method is effective.

**Weaknesses:**

1.	Incremental Contribution: As mentioned before, the core innovation here is very thin. It is essentially adding a "zero-cost" gate to the existing LoRA framework, making it an incremental combination and optimization of existing technologies (LoRA, MoE).

2.	Metaphor Misuse ("Mini-Expert"): The paper repeatedly calls rank-1 components "mini-experts." This is a significant conceptual overstatement. In a standard MoE, an "expert" is typically a full FN layer with complex non-linear transformation capabilities. A rank-1 matrix, however, merely defines a single update direction in a high-dimensional space. What Gated LoRA does is, more accurately, "dynamic selection of feature directions" within LoRA's low-rank subspace, not routing between multiple complex "reasoning paths." This misuse of metaphor is misleading.

3.	ntroduction of New Complexity (Hyperparameter Complexity): While claiming "no extra parameters," the method introduces new hyperparameters. Gated LoRA's training and tuning complexity is actually increased. The paper does not discuss the sensitivity of these new hyperparameters and only provides fixed values.

4.	Ignoring Training Overhead: As mentioned earlier, this method (especially the ReLU variant) trades significantly increased training time for performance gains. For many resource-constrained scenarios, longer training time can be a more significant issue than a few extra parameters.

**Questions:**

1.	On the definition of "expert": Can the authors provide a more rigorous argument for why a rank-1 matrix can be called an "expert"? What is its functional comparability (beyond just "being selected") to a full FFN expert in an MoE?

2.	On training overhead: The training time for the ReLU gate (Table 1, 1.26s/batch) is more than double that of standard LoRA (0.61s/batch). Do the authors consider this overhead reasonable? If I were to double the rank r of standard LoRA (which would also increase training time), could it achieve performance comparable to Gated LoRA (r=128, k=32)?

3.	How were these regularization coefficients chosen? Were they fixed at 1.0 for all models and datasets? How would performance change if these values were altered (e.g., 0.1, 0.01)? This is crucial for assessing the method's ease of use.

---

> ### Author Response · Authors · 2025-12-01
>
> # Training Overhead
>
> The reviewer wrote that "training time for the ReLU gate (Table 1, 1.26s/batch) is more than double that of standard LoRA (0.61s/batch)." This is **factually incorrect** due to a **misreading of Table 1**: In Table 1, 1.26 s/batch corresponds to **inference time**, not training. For LLaMA-3.1-8B multi-task, the training and inference times are:
>
> **Table A.** Comparison of sec/batch latency during training and inference.
> | Method | Train. sec/batch | Infer. sec/batch |
> | ---------- | --------------------- | --------------------- |
> | LoRA | 0.61 | 1.22 |
> | LoRAMoE | 1.60 (x2.62) | 3.09 (x2.53)  |
> | MoLA | 1.42 (x2.32) | 1.92 (x1.57) |
> | Gated LoRA (top-32) | 0.64 (x1.05) | 1.33 (x1.09) |
> | Gated LoRA (ReLU) | 0.68 (x1.11) | 1.26 (x1.03) |
>
> Gated LoRA increases training time by only \~11% and inference by \~3-9% relative to LoRA, where training and inference slowdowns are in the x2~3 range.
>
> # Doubling Rank
>
> The reviewer asks whether doubling the rank of standard LoRA could match Gated LoRA. This question is directly addressed by our **Appendix D** ablations.
>
> In the ablation, with total rank r=128, Gated LoRA with selective activation of k=32 ranks achieves 82.5% average accuracy, the highest among the r=128 configurations we tested. Standard LoRA with r=128 (all ranks activated) achieves 80.9%. In other words, simply adding capacity (more ranks) without selectivity does not match Gated LoRA's performance at the same total rank.
>
> This indicates that the benefit comes from selectivity, rather than just increasing the number of parameters. Gated LoRA uses the same rank budget as LoRA but allocates it more effectively via dynamic rank selection.
>
> # "Mini-Expert" Metaphor
>
> The reviewer argues that calling rank-1 components "mini-experts" is a conceptual overstatement. We agree with the narrow point that a rank-1 factor is not equivalent to a full FFN expert, and in the revised manuscript, we therefore avoid using "mini-expert" as a technical term. Instead, we describe each rank-1 component as a **feature direction** within LoRA’s low-rank subspace and refer to our method as **rank-wise gated LoRA**.
>
> At the same time, we would like to clarify that in MoE-based PEFT (e.g., LoRAMoE, MoLA), "expert" are often LoRA adapters or small adapter modules added in parallel to a shared base model, not full FFN layers. In this literature, "expert" generally refers to any parameterized submodule that is selected conditionally by a router, not necessarily to a large, standalone FFN. Moreover, the notion of experts as "multiple complex reasoning paths" made by the reviewer is itself metaphorical and not part of the standard MoE definition.
>
> Gated LoRA follows the same structural principle of **conditional computation**-- input-dependent selection over a set of subcomponents -- but applies it at the level of rank-1 feature directions within a single LoRA adapter, rather than over full adapters or FFN blocks. Our original use of "mini-expert" was intended as an informal analogy to this conditional selection, not as a claim that each rank-1 direction is comparable in capacity to a full FFN. To avoid any potential confusion, we now refrain from using "mini-expert" in the technical description and instead adopt the reviewer’s more precise wording: **dynamic selection of feature directions in a low-rank subspace.**
>
> # Incremental Contribution
>
> We respectfully disagree with characterizing our contribution as "thin" or "incremental combination." Our ablation studies (**Tables 3-4**) show that the **dual-purpose design** is not equivalent to simply "adding a gate" on top of LoRA.
>
> Concretely, we compared Gated LoRA to a variant that uses a **separate gating projection** (i.e., an extra matrix for gating distinct from the LoRA down-projection). The dual-purpose design achieves the same or higher performance **while using fewer parameters** than the separate-gating variant.
>
> Our goal is precisely to show that a **straightforward modification**, reusing $\mathbf{A}x$ for both adaptation and gating with no new trainable parameters, can approximate some benefits of MoE-style conditional computation. We believe the attractiveness of Gated LoRA lies in this simplicity: it offers consistent gains over LoRA and MoE-PEFT baselines at **zero additional parameter overhead and small computational cost**, which is valuable in practice even if the architectural change is conceptually light.
>
> # Hyperparameter Sensitivity
>
> We used $\lambda_1=\lambda_2=1.0$ across all experiments with consistent results. We acknowledge that a more comprehensive sensitivity analysis would strengthen the work, though we note that LoRA itself is known to be inherently sensitive to hyperparameters such as $\alpha$, $r$, and the learning rate [1,2].
>
> [1] LoRA Learns Less and Forgets Less. TMLR'24.
>
> [2] DoRA: Weight-Decomposed Low-Rank Adaptation. ICML'24.

---

### Official Review · Reviewer_mL7Q · 2025-10-31

**Soundness:** 2
**Presentation:** 3
**Contribution:** 2
**Rating:** 4
**Confidence:** 4

**Summary:**

This paper introduces Gated LoRA, a new PEFT method to reduce task interference in standard LoRA during multi-task fine-tuning. It treats each rank-1 LoRA as a small expert and uses an input-dependent gate (top-k or ReLU) to activate them sparsely. The method’s core design is a dual-use projection that reuses LoRA’s A matrix to produce both features and gating signals, adding no extra parameters.

**Strengths:**

- The paper is well written and easy to follow.

- The design that reuses the A matrix for gating is clever and elegant.

**Weaknesses:**

- The improvement in Table 1 is quite small. It is unclear whether the gain is due to the proposed method or just random variation.

- In some settings, increasing the LoRA parameters even hurts performance. The paper does not show whether the method works reliably across different ranks, especially for low-rank cases.

- There are no examples showing how or why multi-task LoRA fine-tuning causes task interference. Would increasing the rank reduce this interference? And why does multi-task learning lead to interference instead of mutual improvement? This seems a bit counterintuitive.

**Questions:**

Please see weaknesses.

---

> ### Author Response · Authors · 2025-12-01
>
> # Magnitude of Improvement
>
> The reviewer is concerned that the improvements in Table 1 are small and might be random variation. In the revised manuscript, we further evaluate our method on **Phi-3** and **Qwen-2**  backbones, shown in Table A.
>
> **Table A:** Accuracy on multi-task fine-tuning with diverse backbones.
> | Backbone | Method | avg. (%) | gain (%p) |
> | --------------| ---------- | ------ | ------ |
> | Phi-3-Mini-4K-Instruct | LoRA | 82.7 | - |
> | | LoRAMoE | 83.7 | +1.0 |
> | | MoLA | 83.4 | +0.7 |
> | | Gated LoRA (top-32) | **84.2** | **+1.5** |
> | | Gated LoRA (ReLU) | 83.7 | +1.0 |
> | |
> | Qwen2-7B | LoRA | 83.0 | - |
> | | LoRAMoE | 85.6 | +2.6 |
> | | MoLA | 84.8 | +1.8 |
> | | Gated LoRA (top-32) | **86.2** | **+3.2** |
> | | Gated LoRA (ReLU) | 86.0 | +3.0 |
>
> Together with our LLaMA-2-7B/13B and LLaMA-3.1-8B results in the main tables, this gives **five** different backbones. Across all of them, Gated LoRA improves over standard LoRA in both multi-task and single-task settings, and often matches or outperforms MoE-PEFT baselines despite using fewer trainable parameters. We believe that this **consistent pattern of gains across model families and tasks** strongly suggests that the improvements are not due to random noise or a single favorable configuration.
>
> # Rank Robustness
>
> The reviewer stated that the paper does not show whether the method works reliably across different ranks, especially in low-rank cases. This is not true. We respectfully point out that the original submission **already contained r/k ablations in Appendix D**,
>
> Specifically, we tested:
> - Total ranks: $r \in \{ 32, 64, 128 \}$
> - Active ranks: $k \in \{8, 16, 32, 64, 128 \}$
> - 12 configurations for top-k gating (Table 9)
> - 12 configurations for ReLU gating (Table 10)
>
> These experiments show that:
> 1. At fixed r, sparse activation ($k < r$) consistently outperforms dense usage ($k=r$), and
> 2. Performance advantage holds across low-rank regimes  (e.g., r=32, k=8  achieves 83.1% vs r=32, k=32 at 80.3%)
>
> We believe this extensive empirical evidence, already present in the original submission, directly addresses the reviewer's concern about rank reliability.
>
> #  Task Interference
>
> The reviewer asks for concrete evidence that multi-task LoRA induces interference and questions whether simply increasing rank would avoid it.
>
> First, multi-task LoRA interference is well-documented in prior work. **Prior work** has extensively documented task interference in multi-task LoRA-style settings, and we cited several such works. In the revision, we additionally included  MTL-LoRA [1], which visually demonstrates that projecting multiple task-specific features into the same dense low-rank space leads to interference.
>
> Beyond prior work, our own results also support this picture. Comparing the main results in **Table 1** (multi-task) and **Table 2** (single-task), we find several benchmarks where multi-task LoRA underperforms single-task LoRA on specific tasks, indicating that sharing a single low-rank subspace can hurt individual tasks rather than yielding mutual improvement.
>
> Furthermore, from **Appendix D**, standard LoRA with r=128 (all rank active) shows **80.9%** average accuracy, while **Gated LoRA** with r=128, k=32 (selective) shows **82.5%** average accuracy, showing that simply adding capacity (more ranks) without selectivity does not solve the interference. In fact, our ablations show that using all 128 ranks performs worse than selectively using 32 of them. The key insight is that interference is reduced not by more parameters, but by enabling different inputs to activate different rank subsets.
>
> Our key insight is that when all ranks are always active, gradients from different tasks compete for the same directions. When gradients from different tasks are not aligned, these shared directions become a source of conflict: updates that help one task can push another task in a harmful direction. Selective gating allows different inputs to use different rank subsets, reducing this competition. This is the core motivation for our method, and the main results and ablations support our intuition.
>
> [1] MTL-LoRA: Low-Rank Adaptation for Multi-Task Learning. AAAI'25.

---

### Official Review · Reviewer_Jqp3 · 2025-11-01

**Soundness:** 2
**Presentation:** 2
**Contribution:** 2
**Rating:** 2
**Confidence:** 3

**Summary:**

This paper proposes Gated LoRA, an extension of Low-Rank Adaptation (LoRA) for parameter-efficient fine-tuning of LLMs. Instead of uniformly activating all LoRA rank directions, the authors interpret each rank-1 component as a mini-expert and introduce input-dependent gating to select only relevant ranks per token.
A key design is the dual-purpose projection — the same LoRA down-projection matrix A is reused both for feature extraction and gating, avoiding additional routing parameters.

Two selectors are explored: fixed top-k gating and ReLU gating with sparsity regularization.
Experiments on nine NLU benchmarks using LLaMA backbones show moderate improvements (up to +3.6 points) over standard LoRA while keeping training efficiency nearly identical.

**Strengths:**

The idea of reusing LoRA’s down-projection for gating (dual-purpose projection) is elegant and avoids the parameter overhead common in MoE-style PEFT methods.

The proposed gating mechanism achieves conditional activation without introducing additional routing layers or large routers.

**Weaknesses:**

[Misleading “MoE-like” terminology]
The paper repeatedly refers to the method as “MoE-style”, yet no actual expert routing or token-level dispatch exists. Gating operates purely within the LoRA rank dimensions (dimension-level gating), not across separate expert modules as in true MoE architectures. The term “mini-expert” is metaphorical rather than architectural.

[No additional memory efficiency beyond LoRA]
The paper’s title and abstract suggest improved “parameter-efficiency” or “memory-efficiency.” In reality, trainable parameters remain identical to standard LoRA; the only efficiency stems from activation sparsity, not parameter reduction. Thus, “memory-efficient” is misleading.

[Limited experimental diversity]
All experiments are confined to LLaMA-family models (LLaMA-2-7B/13B and LLaMA-3.1-8B). There is no evaluation on other architectures (e.g., Mistral, Falcon, OPT, or encoder-only models like RoBERTa), leaving the general applicability unverified.

[Marginal computational benefit]
The reported inference latency gain (≈0.04–0.1s per batch) is minor and may fall within measurement noise.
Claims of efficiency would be stronger with FLOP analysis or hardware-level profiling.

**Questions:**

Avoid using “MoE-like” unless token-level routing or explicit expert partitioning is introduced. More accurate terms would be “Rank-wise gated LoRA” or “Self-gated LoRA.”

Include experiments on non-LLaMA architectures or smaller encoder-only models to support claims of generality.

---

> ### Author Response · Authors · 2025-12-01
>
> We thank R1 for the detailed feedback. We address each concern below and note that while we respectfully disagree with some interpretations, we have substantially revised the manuscript to clarify these points.
>
> # "MoE-like" Terminology and Architecture
>
> The reviewer argues that the paper "repeatedly refers to the method as ‘MoE-style’" and suggests it is misleading because there is no true MoE routing or separate experts. This is not an accurate description of our wording.
>
> In the original submission, we never used "MoE-style" to describe Gated LoRA itself. The phrase "MoE-style" is used to describe **existing PEFT baselines** such as LoRAMoE and MoLA, which explicitly adopt a Mixture-of-Experts design with separate experts and routers. For our own method, we consistently used phrases like "inspired by MoE-style conditional computation" or "MoE-inspired sparse activation over LoRA ranks" to indicate conceptual motivation while clearly differentiating from traditional MoE architectures.
>
> We also want to clarify the reviewer’s statement that "no token-level routing" is involved. While it is true that we do not perform routing between separate expert modules, our gating is still token-level. Gated LoRA performs token-wise gating over rank directions within a single adapter, rather than token-wise routing across multiple experts as in standard MoE.
>
> To avoid any further misunderstanding, in the revised manuscript, we make this distinction even more explicit. We describe our method as **rank-wise gated LoRA**, and rephrase the language around "mini-experts" to emphasize that rank-1 components are **feature directions** in a low-rank subspace, not experts comparable to full FFN blocks in classical MoE. We hope this clarifies that our use of "MoE-style" in the paper is limited to prior PEFT baselines, and that Gated LoRA itself is positioned as a rank-wise gating mechanism.
>
> # Parameter / Memory Efficiency
>
> The reviewer stated that the title and abstract suggest improved "parameter-efficiency" or "memory-efficiency" over LoRA, while trainable parameters remain identical, and thus "memory-efficient is misleading." This misinterprets our claims.
>
> Our actual parameter-efficiency claim was that it uses **exactly the same trainable parameter count as standard LoRA,** while MoE-PEFT baselines such as LoRAMoE and MoLA introduce additional adapters and routing parameters. In other words, Gated LoRA achieves LoRA-level parameter efficiency while adding input-dependent rank selection, and remains substantially more parameter-efficient than MoE-PEFT methods.
>
> Moreover, the phrase "Parameter-Efficient Fine-Tuning  (PEFT)" in our title refers to the **established class of fine-tuning methods** that use fewer parameters than full fine-tuning. It is not intended to assert that our method is more parameter-efficient than LoRA itself. We believe this clarification directly addresses the reviewer’s concern: we are not claiming improved parameter count over LoRA, only that we maintain LoRA-level parameters while adding input-dependent rank selection.
>
> # Experimental Diversity
>
> The reviewer noted that our original experiments were confined to LLaMA-family models and requested evidence from other architectures. In the revised manuscript, we have added multi-task experiments on two additional decoder-only backbones: **Phi-3** and **Qwen-2**. We summarize our results in **Table A:**
>
> **Table A:** Accuracy on multi-task fine-tuning with diverse backbones.
> | Backbone | Method | avg. (%) | gain (%p) |
> | --------------| ---------- | ------ | ------ |
> | Phi-3-Mini-4K-Instruct | LoRA | 82.7 | - |
> | | LoRAMoE | 83.7 | +1.0 |
> | | MoLA | 83.4 | +0.7 |
> | | Gated LoRA (top-32) | **84.2** | **+1.5** |
> | | Gated LoRA (ReLU) | 83.7 | +1.0 |
> | |
> | Qwen2-7B | LoRA | 83.0 | - |
> | | LoRAMoE | 85.6 | +2.6 |
> | | MoLA | 84.8 | +1.8 |
> | | Gated LoRA (top-32) | **86.2** | **+3.2** |
> | | Gated LoRA (ReLU) | 86.0 | +3.0 |
>
> Under the same five-task multi-task setting, Gated LoRA consistently improves over standard LoRA and outperforms MoE-PEFT baselines. These new results are reported in the appendix and referenced in the main text, and they demonstrate that the method is not tied to LLaMA-specific architectural details.
>
> # Computational Benefit
>
> The reviewer commented that the reported latency "gains" of ≈0.04–0.1s per batch is too small. This is precisely our intended advantage. Our primary efficiency claim is that Gated LoRA preserves LoRA's trainable parameter budget, incurs only a small computational overhead compared to LoRA, and is much cheaper than MoE-PEFT baselines. The small latency gain (or increase) actually implies minimal computational overhead, especially compared to LoRAMoE and MoLA, which show a latency increase of 0.7~1.9s.

---

### Author Response · Authors · 2025-12-01

We thank the reviewers for their constructive feedback. In response, we have uploaded a revised manuscript with significant improvements.  We also respectfully correct key factual errors present in reviews regarding our training overhead and benchmark performances.

**1. Terminology Refinement.** We have revised the paper to remove the term "mini-expert" and "MoE-style" to avoid confusion. We now use the more precise terms **rank-wise gated LoRA** and "fine-grained feature directions" to accurately describe our dimension-level gating mechanism.

**2. Expanded Experimental Scope.**  To address concerns about generalization beyond LLaMA, we added Appendix E, which evaluates Gated LoRA on Phi-3-Mini-4k and Qwen2-7B. Gated LoRA outperforms standard LoRA and MoE baselines on these diverse architectures (e.g., +3.2% gain on Qwen2-7B vs LoRA), confirming that our gains are architectural, not model-specific.

**3. Factual Corrections on Efficiency & Performance.**
- **Training Overhead:** Reviewer j89t incorrectly cited our inference latency (1.26s) as training time. The actual training latency is only 0.68s, not the "2x" claimed compared to the standard LoRA (0.61s).
- **Reasoning Performance:** Reviewer kaxG claimed we underperform on GSM8K. This overlooks our ReLU variant, which outperforms all baselines for both multi-task and single-task fine-tuning.

---

### Meta-Review · Area_Chair_VBVZ · 2026-01-05

**Summary:**

This paper proposes Gated LoRA, a PEFT method that performs token-wise gating over LoRA rank directions. The key idea is a dual-purpose projection: the LoRA down-projection matrix  $A$ is reused to produce both low-rank features and gating signals, enabling input-dependent sparse activation with no additional trainable parameters beyond standard LoRA. Two gating variants (top-$k$ and ReLU) are explored, along with regularization to encourage sparsity/orthogonality. Reviewers agree the design is simple and practical and that empirical gains over LoRA/MoE-PEFT baselines are generally positive, but they raised concerns about novelty framing, terminology clarity, breadth of evaluation, and the interpretation of efficiency results. The authors provided a rebuttal with a revised manuscript, including terminology refinements, additional backbone experiments (Phi-3, Qwen2), and corrections to factual errors regarding latency measurements and benchmark interpretation.

**Reviewer Concerns:**

### Concerns addressed by the rebuttal:

Terminology and framing: The authors revised language to avoid misleading “MoE-like” or “mini-expert” interpretations, clearly positioning the method as rank-wise gated LoRA.

Model diversity: Additional experiments on Phi-3 and Qwen2-7B address concerns about LLaMA-only evaluation.

Efficiency clarification: Reviewers’ confusion between training and inference latency was corrected; training overhead is shown to be close to LoRA and substantially lower than MoE-PEFT baselines.

### Remain concerns:

Incremental novelty: While elegant, the method remains a lightweight architectural extension of LoRA; some reviewers may still view the contribution as incremental.

Statistical robustness: Gains are moderate, and reviewers asked for stronger evidence via repeated runs or significance analysis.

Cost characterization: Although clarified, efficiency claims would benefit from more explicit FLOP or profiling analysis and clearer discussion of added hyperparameter tuning.

**Reviewer Scores:**

I think other reviewers should keep the score and
Jqp3: 2 → 4 (key concerns addressed, novelty still modest),

kaxG: 2 → 4 (architecture diversity and GSM8K clarification help, but novelty concerns remain).

---

### Decision · Program_Chairs · 2026-01-26

Reject